# 2-iminobiotin, a selective inhibitor of nitric oxide synthase, improves memory and learning in a rat model after four vessel occlusion, mimicking cardiac arrest

**Cacha Peeters-Scholte**[1]*, **Sigal Meilin**[2], **Yafit Berckovich**[2], **Paul Westers**[3]

**1** Neurophyxia B.V., Den Bosch, The Netherlands, **2** Neurology Service, MD Biosciences Ltd, Nes-Ziona, Israel, **3** Julius Center for Health Sciences and Primary Care, University Medical Center Utrecht, Utrecht, The Netherlands

* cacha.peeters@neurophyxia.com

**Data Availability Statement:** The data underlying the results presented in the study are available from https://www.neurophyxia.com/en/literature/.

## Abstract

Survivors of out-of-hospital cardiac arrest (OHCA) experience between 30% and 50% cognitive deficits several years post-discharge. Especially spatial memory is affected due to ischemia-induced neuronal damage in the hippocampus. Aim of this study was to investigate the potential neuroprotective effect of 2-iminobiotin (2-IB), a biotin analogue, on memory and learning in a four-vessel occlusion model of global ischemia using the Water Maze test. Sprague-Dawley rats were randomly assigned to either sham operation (n = 6), vehicle treatment (n = 20), 1.1 (n = 15), 3.3 (n = 14), 10 (n = 14), or 30 mg/kg/dose 2-IB treatment (n = 15). Treatment was subcutaneously (*s.c.*) administered immediately upon reperfusion, at 12h, and at 24h after reperfusion. Memory function on day 32 was significantly preserved in all doses of 2-IB rats compared to vehicle, as was the learning curve in the 1.1, 3.3 and 30 mg/kg dose group. Adult rats treated *s.c.* with 3 gifts of 2-IB every 12 h in a dose range of 1.1–30 mg/kg/dose directly upon reperfusion showed significant improved memory and learning after four vessel occlusion compared to vehicle-treated rats. Since 2-IB has already shown to be safe in a phase 1 clinical trial in adult human volunteers, it is a suitable candidate for translation to a human phase 2 study after OHCA.

## Introduction

Ischemic heart disease, including the occurrence of out-of-hospital cardiac arrest (OHCA), is the leading cause of mortality worldwide [1]. With the increasing awareness of basic life support and the nearby presence of automated external defibrillators, survival rates after OHCA tend to increase the past decades [2]. For that reason, the consequences on cognitive impairment, quality of life, and effects of everyday life functioning become more and more important [3].

In OHCA survivors 30–50% experience cognitive deficits up to several years post-discharge [4]. Even among patients with good neurological recovery at hospital discharge (with a

**Funding:** This study was sponsored by Neurophyxia B.V., 's Hertogenbosch, the Netherlands. CPS is a consultant for and stockholder of Neurophyxia and has received honoraria from Neurophyxia for consultancy work; SM and YB are employees of MD Biosciences Ltd., Israel, which is a contract research organization which received funding from Neurophyxia to perform the experiments described herein.

**Competing interests:** I have read the journal's policy and the authors of this manuscript have the following competing interests: CPS has stocks of Neurophyxia BV and is co-inventor onthe patents. Neurophyxia BV holds several patent families: 2-iminobiotin formulations and uses thereof (WO2011149349), 2-iminobiotin for use in the treatment of brain cell injury (WO2017105237); 2-iminobiotin for use in the treatment of stroke (WO2022203504 and WO2022203505).

Cerebral Performance Category Scale≤2), 29% experience memory problems [5] and 43% cognitive impairment [6].

To improve survival and decrease the degree of neurological impairment, targeted temperature management (TTM) after OHCA has been implemented. No difference in cognitive function was demonstrated between patients that received hypothermia (32–34˚C) and controlled normothermia (36˚C) in the TTM1 trial [7]. The results of the TTM2 trial are awaited (ClinicalTrials.gov, NCT03543371).

New strategies are being explored to reduce the consequences on cognition after OHCA.

Besides global measures to reduce cell damage post cardiac arrest, past years interest has been gained for the role of nitic oxide in brain cell injury [8]. Especially the use of selective inhibitors of the inducible and neuronal form of nitric oxide synthase (NOS) have been described as potential rescue agents following hypoxic-ischemic encephalopathy [9]. Inhibition of NOS by 2-iminobiotin (2-IB), a biotin analogue, reduced the amount of neurological damage in a piglet model of perinatal global hypoxia-ischemia (HI) [10]. 2-IB has been tested for safety and tolerability in several clinical Phase 2 trials for this indication in neonates [11, 12]. Although the defence mechanisms after global ischemia differ between neonates and adults, it was hypothesized that 2-IB could also have beneficial effects on neurological outcome, especially on memory, after HI reperfusion injury in adults.

The modified four-vessel (4VO) occlusion model in adult rats is an effective model to study the consequences of transient but severe brain ischemia [13]. This procedure leads to a specific injury to the hippocampus and a subsequent decline in cognitive performance. In the current study we used the 4VO model to investigate whether 2-IB, administered upon reperfusion, can improve cognitive function in a 4VO rat model, mimicking cardiac arrest in adults.

## Methods

### Animals

All animal procedures were performed in accordance with the Guide for the Care and Use of Laboratory Animals published by the National Institutes of Health, and were approved by the Animal Care and Use Committee of the University of Israel (MD-3-1-195-1540). All attempts were made to maximize animal welfare while preserving scientific validity and integrity. A humane endpoint was defined for animals with a poor prognosis of quality of life, or severe distress. All efforts were made to minimize suffering; when needed, rats were then euthanized with an overdose pentobarbital.

In this model healthy, test-naïve, female Sprague-Dawley rats of eight to nine week old, and weighing about 200 gram were used (Envigo, Israel). Weight variation of the animals at time of treatment initiation did not exceed 20% of the mean weight.

Animals were given a unique animal identification tail mark. Rats were allowed a minimum of 5 days to acclimate to their surroundings and were provided food (a commercial, sterile rodent diet, Envigo, Israel) and water ad libitum. During the acclimatization and study period rats were housed in limited access rodent facilities and were kept in groups with a maximum of 4 rats per cage. The cages were made of polypropylene, fitted with solid bottoms and filled with sterile wood shavings as bedding material.

Automatically controlled environmental conditions were set to maintain temperature at 17–23˚C with a relative humidity of 30–70%, a 12:12 hour light:dark cycle and 15–30 air changes/h in the study room. Temperature and relative humidity was monitored daily and the light cycle was monitored by a control clock. Temperature was controlled during occlusion and reperfusion.

## Four-vessel occlusion model

Rats were anesthetized using ketamine (90 mg/kg) and xylazine sodium (10%), as described earlier [14]. In short, the first cervical vertebrata was exposed and the vertebral arteries were permanently occluded by electrocauterization. Twenty-four hours post vertebral arterial occlusion, the animals were re-anesthetized using isoflurane. The common carotid arteries were exposed and occluded using microaneurysm clips for 15 minutes. After removing the clips reperfusion occurred. The skin of the incision at the neck was sutured using 4–0 silk suture.

During the operation animal's core temperature was monitored using a rectal probe (Model 400; YSI Inc., Yellow Springs, OH, USA) connected to a thermometer (Model 8402–00; Cole-Parmer Instrument Co. Ltd, London, UK). The ischemic insult was initiated when a rectal temperature of 37–38˚C was achieved, and this temperature was maintained throughout the procedure using a heating mattress.

The presence of global ischemia was verified by removing the anesthesia for a few seconds during ischemia. Animals that did not show a righting reflex were considered ischemic, and were included in the study. Animals that showed unusual behavior not characteristic of historical behavior in this model at 24 hours post ischemia were excluded from the study and were sacrificed in a human way.

## Study design

The study included 8 phases: 1. acclimatisation phase; 2. preparation phase; 3. HI phase 4. treatment phase; 5. follow-up phase; 6. learning test; 7. memory test; 8. termination phase (for overview, see Fig 1). During the acclimatisation period a total of 106 rats were randomly assigned to treatment groups (n = 20 per group) or sham operation (n = 6). The sham-operated rats were subjected to anesthesia, but no arteries were occluded. The remaining 100 rats received permanent vertebral occlusion, and after 24h both common carotid arteries were transiently occluded for 15 minutes. Seventy-five rats were considered ischemic and were assigned to treatment groups using a randomization table. Vehicle or 2-IB was administered *s. c.* directly upon reperfusion, at 12h and at 24h after reperfusion. Body weight was measured weekly. Blood glucose values were measured at baseline and at day 1 and 2 after reperfusion. From day 3 until day 27 rats were monitored periodically for any abnormalities. From day 28 until day 31 the learning test was performed. The memory test was performed on study day 32. Rats were sacrificed on day 33 with an overdose of pentobarbital (> 100 mg/kg *i.p.*) and their brains were harvested for further histological analysis.

## Treatment

Rats were either treated with vehicle (n = 20), 1.1 mg/kg/dose 2-IB (n = 15), 3.3 mg/kg/dose 2-IB (n = 14), 10 mg/kg/dose 2-IB (n = 13) or 30 mg/kg/dose 2-IB (n = 13), dissolved in 5 ml/

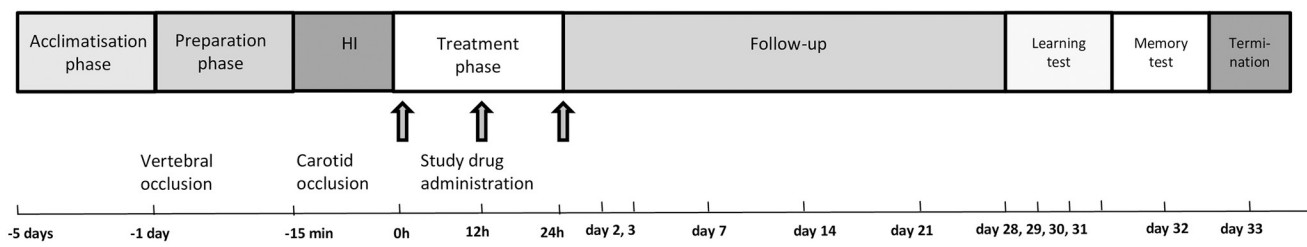

**Fig 1. Overview of study design.** HI = hypoxia-ischemia;

kg normal saline at pH 3.8–4.0. Animals were *s.c.* dosed immediately upon reperfusion, at 12h and at 24h after reperfusion, a treatment regimen already demonstrated to be neuroprotective in a neonatal HI rat model with an optimal dose regimen of 10 mg/kg/dose [15]. Each dosing group was kept in separate cages to avoid cross-contamination by consuming stools during the study period.

### Morris water maze test

Tests were performed during day-time. Rats were introduced into a standardized 1.2 m-diameter pool filled with water for 60 sec or until they located a platform hidden 1 cm below the water surface. Several visual cues were provided within the room in which the pool was located to allow rats to spatially navigate the water maze. Rats that located the hidden platform were allowed to remain on it for 10 sec, and rats that failed to find the platform within 120 sec were placed on the platform for 10 sec. Rats were allowed two attempts to find the hidden platform, and this learning test was performed over a period of 4 days (days 28–31).

The memory test was performed on the fifth day (day 32), at which time the hidden platform was removed, and rats were placed in the pool for a single 60-sec. trial. The amount of time that each rat spent in the quadrant, where the hidden platform had previously been located, was recorded by an observer who was blinded to the experimental groups. The Morris Water Maze (MWM) was used to evaluate how well the rats remembered the location of the hidden platform, and whether they had learned to navigate towards the appropriate quadrant. The time that the rat spent in the right quadrant was measured. This was considered the primary outcome.

During the learning test the time to find a hidden platform in the water was also calculated, as well as the area under the curve (AUC) of the subsequent learning tests. These parameters were considered secondary outcomes.

### Histology

Histological analysis was performed in the remaining rats for each treatment group and in 3 sham-operated rats. The brains were perfused via the left ventricle with heparinised normal saline to remove the excess of blood from the brain. Afterwards the brain was perfused with 4% paraformaldehyde phosphate-buffered saline. The whole brain was removed and immersed in formaldehyde for at least 72h. Paraffin embedded coronal sections (6mm) were cut at approximately 3.3mm posterior to the bregma and stained with hematoxylin–eosin. Sections were scored in a blinded way for living neurons in the right and left part of the CA1 region of the hippocampus, the most affected area in this model of global ischemia.

### Statistical analysis

The sample size of the primary outcome parameter, the memory test, was determined based on the two independent sample t-test with a power of 80%, and an estimated mean difference of 8 (SD = 4) with Bonferroni correction for multiple comparisons. At least 13 eligible rats per treatment group were needed. The learning test and histology were secondary outcome parameters.

All data is expressed as mean ± SEM. Statistical evaluation of the data was performed using one-way ANOVA when appropriate. For variables changes over time, repeated measurement analysis was used with baseline as co-variate for glucose and bodyweight. Post-hoc analysis was performed using Dunnett's two-sided test with vehicle group as there reference. A p-value smaller than 0.05 was considered statistically significant. Statistical analysis was performed using SPSS (IBM, version 23).

## Results

### Animals

A common slight and transient decrease in body weight of all ischemic animals was recorded during the first 2 days following the operation (Fig 2A). There was no significant difference between treatment groups (p = 0.223). Also no significant differences were found in blood glucose values between treatment groups over time (p = 0.423) (Fig 2B).

### Memory test

The memory test, defined as the time spent in the right quadrant on day 32, was 10.1 ± 1.2 sec in sham-operated rats and 3.9 ± 1.0 sec in the vehicle-treated rats. Following treatment with 2-IB (Fig 3A) a significant increase in the time the animals spent in the right quadrant was recorded (10.7 ± 1.1; 12.7 ± 1.8; 12.0 ± 2.0; and 12.5 ± 1.7 sec in 2-IB treated groups with 1.1; 3.3; 10 and 30 mg/kg; p = 0,003, p<0,0005, p = 0,001 and p<0,0005 respectively).

### Learning test

Repeated measurement analysis showed a significant difference between the six groups (p = 0.004) and a significant effect over time (p<0.0005). Dunnett's post hoc analysis with vehicle as reference showed a significant difference for the sham (p = 0.008), the 1.1 mg/kg/dose (p = 0.028), the 3.3 mg/kg/dose (p = 0.009), and the 30 mg/kg/dose (p = 0.023) treated animals (Fig 3B).

Calculation of AUC of the learning test from was 232 ± 34 in sham-operated rats, 349 ± 17 in vehicle-treated rats and 276 ± 22, 267 ± 17, 297 ± 21, 287 ± 23 in the 1.1, 3.3, 10, and 30 mg/kg/dose 2-IB treated rats, respectively. One way ANOVA showed a significant difference between treatment groups (p<0.012). Dunnett's post hoc analysis with vehicle as reference revealed a significant difference for the 1.1 mg/kg/dose (p = 0.040), and 3.3 mg/kg/dose group (p = 0.018).

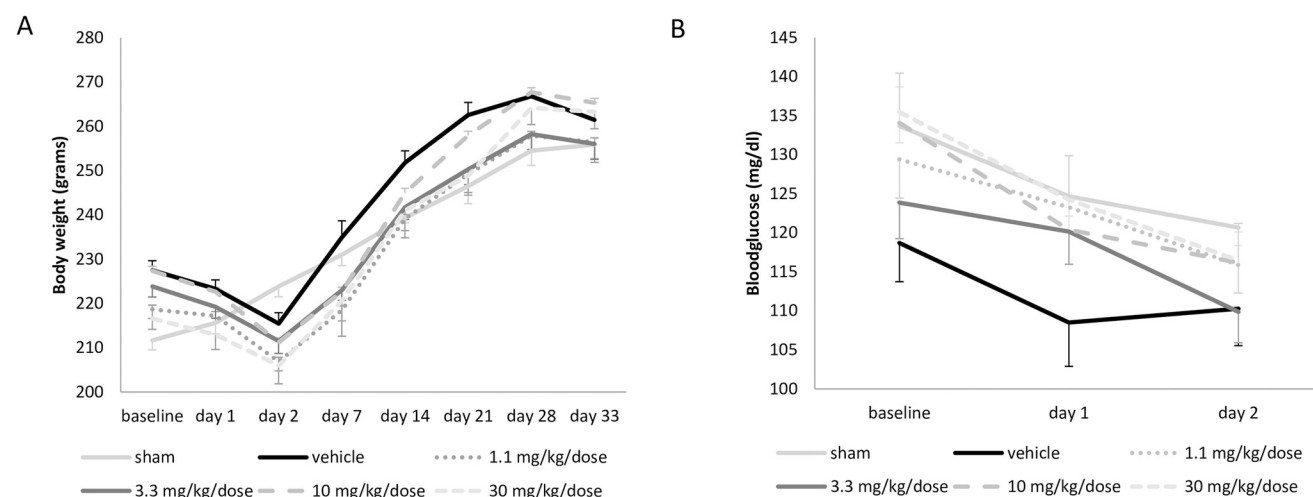

**Fig 2. Body weight and blood glucose.** A. body weight (in grams) and B. blood glucose (in mg/dl) for sham-operated rats (n = 6), vehicle-treated (n = 20), or rats treated with 2-iminobiotin in a dose of 1.1 mg/kg/dose (n = 15), 3.3 mg/kg/dose (n = 14), 10 mg/kg/dose (n = 13) or 30 mg/kg/dose (n = 13) during the study period. Data is shown as mean ± SEM. Dose is given in mg/kg/dose and *s.c.* administered directly upon reperfusion, at 12 and 24h.

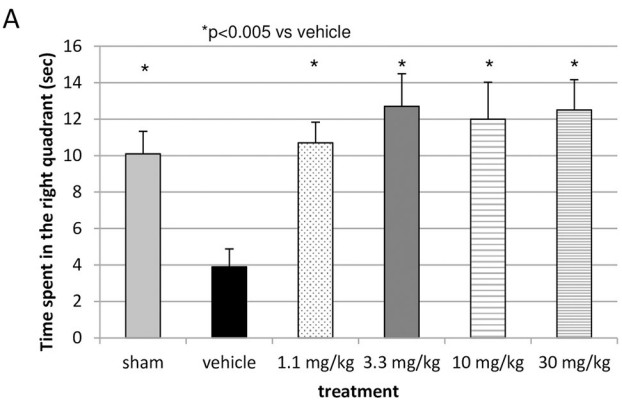
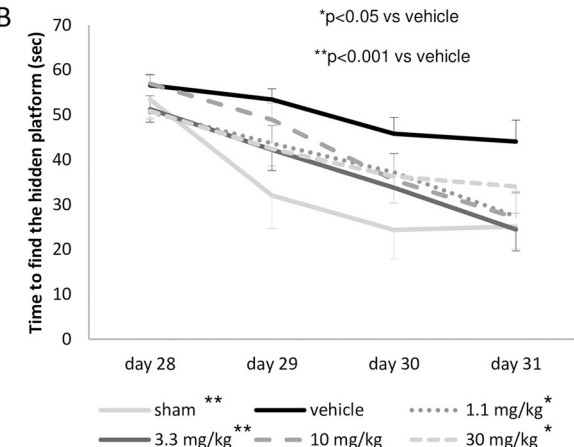

**Fig 3. Memory and learning.** A. Memory test (time spent in the right quadrant to find a hidden platform in seconds (in seconds (mean ± SEM)) at day 32 after 4 vessel occlusion (4VO) and B. learning test from day 28 until day 31 after 4VO: time to find the hidden platform (in seconds (mean ± SEM)) for sham-operated rats (n = 6), vehicle-treated (n = 20), or rats treated with 2-iminobiotin in a dose of 1.1 mg/kg/dose (n = 15), 3.3 mg/kg/dose (n = 14), 10 mg/kg/dose (n = 13) or 30 mg/kg/dose (n = 13) directly upon reperfusion, at 12 and at 24h after reperfusion.

## Histology

Fig 4 shows a representative example of the histology findings in the CA1 region of the hippocampus in a sham-operated (A, B), a vehicle-treated (C, D) and a 3.3 mg/kg 2-IB treated rat (E, F). There was a significant difference in surviving cells in the CA1 region caused by 4VO (one-way-ANOVA p = 0.002), Dunnett's post hoc analysis with sham as reference showed for all other treatments a significant difference (all p<0.020), but not with vehicle as reference. No difference was found between the right and left CA1 region of the hippocampus (S1 Table).

## Discussion

In the present study we investigated the potential neuroprotective effects of 2-IB, a selective inhibitor of nitric oxide synthase, on memory in a 4VO model using the MWM. 2-IB, administered in a dose range of 1.1 to 30 mg/kg/dose upon reperfusion, showed significantly improved memory and learning.

In survivors of OHCA frequently impairments, most often in the memory domain, are seen [16]. Even in patients that were discharged from the hospital with a Cerebral Performance Score of 1 (defined as good cerebral performance), reduced accuracy of working memory and speed of spatial memory was observed [17]. Over the last decade survival of OHCA improved significantly, due to the use of automated external defibrillators, increased bystander cardiopulmonary resuscitation, and improved in-hospital survival [18, 19]. In a recent study it was reported that between 30% to 50% of the patients experience cognitive deficits for up to several years post-OHCA, besides anxiety, depression and fatigue [4]. This makes cognitive impairments and memory problems after OHCA a large social and welfare issue.

The MWM is a procedure in which both the spatial memory (hidden-platform) and the non-spatial (visible platform) conditions of the memory are being tested [20]. This test is being extensively used to measure the effect of potential neuroprotective drugs on neurocognitive disorders affecting spatial learning and memory. In this 4VO model mimicking cardiac arrest, the administration of 2-IB upon reperfusion significantly increased the spatial memory function to a degree comparable to that of the sham condition (Fig 3A).

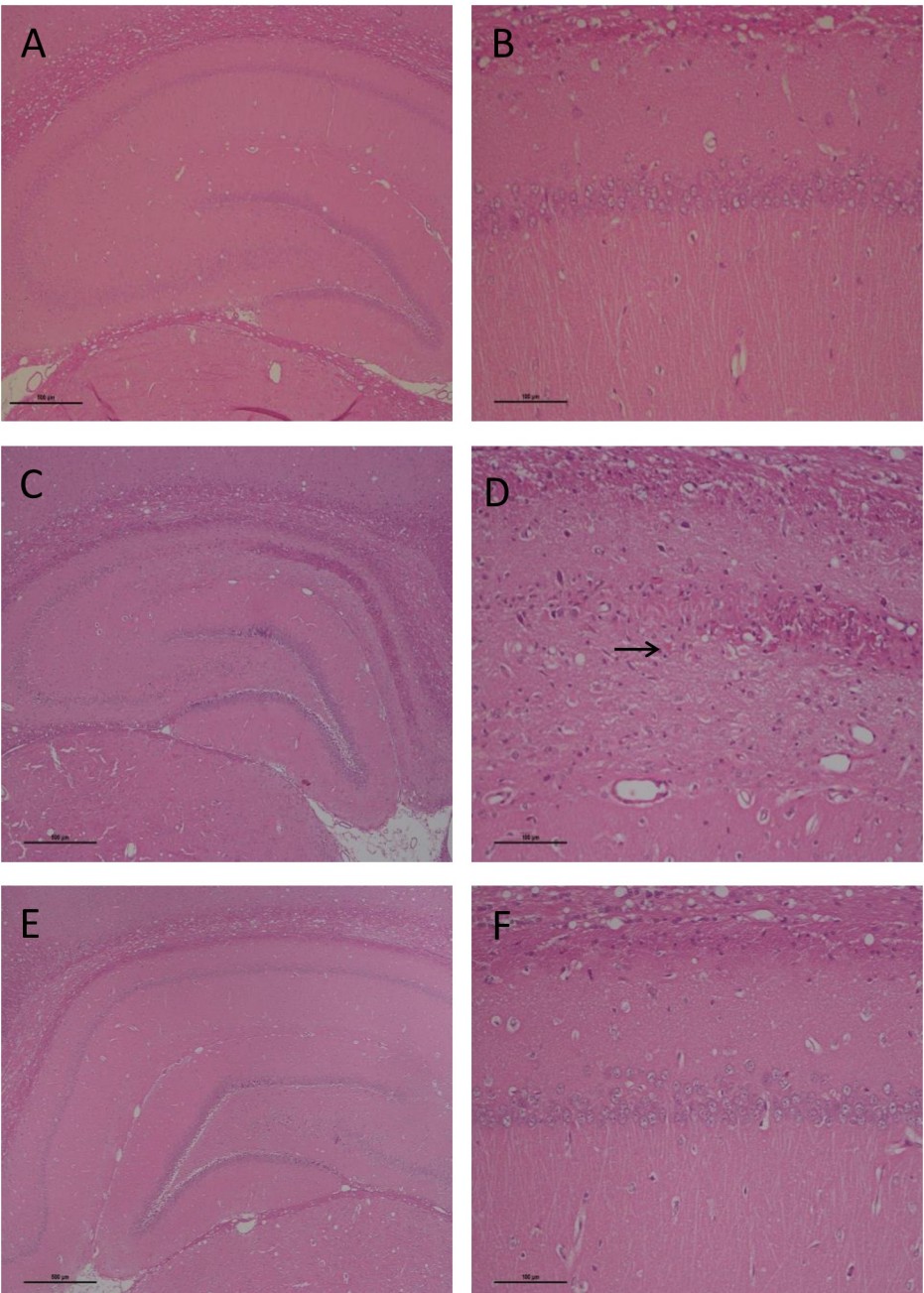

**Fig 4. Histoloy in CA1 region of the hippocampus.** Representative example of the histology of the CA1 region in the hippocampus of a sham-operated rat (A and B), a vehicle treated rat 33 days after 4 vessel occlusion (C and D), and a 3.3 mg/kg/dose 2-iminobiotin treated rat (E and F). Magnification 4x (left images), and 40x (right images). Dead CA1 pyramidal neurons appear as angular dark red triangular structures (see arrow). Gliosis of the neuropil above and below CA1 (outer plexiform and inner polymorph layers, respectively) is seen as increased nuclear density.

In this study only female rats were used. Earlier it was found that differences in MWM performance were found between male and female rodents with male animals showing an advantage in spatial learning [21]. Furthermore, it was shown that 2-IB had neuroprotective effects after HI in p3 and p7 female rats [22, 23], as well as in p7 and p12 rats [15, 24], and in neonatal

HI piglets models of both genders [10, 25]. Gender-specific actions after ischemia have been described both in humans [26], and rats [22, 27]. Also after OHCA, gender differences exist regarding the cause of arrest, adverse events and outcome [28]. In upcoming clinical studies it is very important to sub-analyze the effect of neuroprotective strategies in both genders.

The memory function was assessed 32 days after the 4VO, which is a clinically relevant survival time after OHCA, and brain injury has stabilized. 2-IB treatment was administered only the first day after OHCA, but significant differences in learning and memory between the two groups persisted even after treatment cessation.

In this 4VO model using memory as a primary outcome parameter several compounds [14] and herbs [29–31] have been tested before. Actovegin, an ultrafiltrate derived from calf blood, was administered 6h after 4VO until day 40. Memory was tested at 40 and 68 days, showing a significant effect of actovegin treatment [14]. Renshen Shouwu capsule, a traditional Chinese medicine, given 6 days before until 14 days after 4VO, improved the learning and memory ability of rats 12 days after 4VO [29]. MLC901, also a traditional Chinese medicine, was administered upon reperfusion until 3 to 7 days after 4VO. A preservation in memory function was shown at 12 and 19 days after 4VO [30]. Xiao-Xu-Ming decoction, a Chinese herb cocktail, was given 3 days before 4VO to rats until sacrification. Rats had a significant quicker learning curve at day 8 to 11 after 4VO than vehicle treated rats [31]. More recently, short term effects of paricalcitol were investigated in this model, showing improved neurological function 2h after 4VO and decreased neurodegeneration 4 days after 4VO [32]. Studies that investigate post-ischemia/reperfusion treatment regimens compared to pre-ischemia treatments are more comparable to the clinical situation and might be more relevant when searching for potential treatment strategies following cardiac arrest. Since the protocols differ in treatment schedules and memory is assessed at different time points after 4VO, it remains very difficult to compare the degree of effectiveness of the different treatments after ischemia.

Memory deficits correspond pathophysiologically to neuronal damage mainly in the hippocampus [33]. The 4VO model predominately leads to memory dysfunction based on neuronal injury in the CA1 region of the hippocampus [13]. In this model a spontaneous repopulation occurs in the CA1 region after clinically relevant survival times, but this is not sufficient to offset the behavioral impairments arising from the ischemic insult [34]. In the present study there was a significant decrease in surviving cells in the CA1 region of the hippocampus after HI. Treatment with 2-IB did not show a significant difference versus vehicle in surviving CA-1 cells, probably because repopulation already occurred.

Earlier it was shown in *in vitro* studies, that 2-IB, which has a similar chemical structure as L-Arginine, exhibited selective inhibition of neuronal and inducible nitric oxide synthase activity of murine iNOS and rat cNOS with no effect on endothelial nitric oxide synthase (eNOS) [35]. This was reconfirmed in a radioligand binding assays that demonstrated lower IC50 for inducible NOS(96uM) and neuronal NOS (142 uM) compared to endothelial NOS (646 uM) [36]. This difference in NOS selectivity is important, since endothelial NOS plays an important role in maintaining blood pressure, especially after cerebral HI reperfusion.

What is known about the safety of 2-IB in humans? A Phase 1 clinical trial with 2-IB was carried out in human male volunteers, in which 2-IB showed to be safe and tolerable up to doses of 72 mg/kg/day [37].

In this study 2-IB was administered using different doses in order to find the best dose of 2-IB for future clinical trials. A dose range of 1.1–30 mg/kg/dose, administered *s.c.* every 12h for 3 gifts, showed a significantly increased memory function. The learning test showed a significant difference between vehicle, and the 1.1, 3.3 and 30 mg/kg/dose. When calculating the AUC of the learning test, the 1.1 and 3.3 mg/kg/dose were the most effective. When translating the most optimal *s.c.* dose of 1.1–3.3 mg/kg/dose to exposure in humans, based on data

obtained in healthy volunteers [37], an *i.v.* 2-IB dose of 0.055–0.165 mg/kg/dose every 4 hours for 24 hours, would be optimal for clinical studies after OHCA.

Based on the current proof-of concept data and the earlier demonstrated safety in adult volunteers [37], a Phase 2 trial was performed to evaluate safety and pharmacokinetics in adults after OHCA [38].

Despite the clinical relevance of this in vivo study, limitations associated with the experimental approach should be considered. Since global ischemia might also have an effect on motor performances, an additional assessment of muscle strength and endurance, such as a swim speed test, would be useful to confirm that surgery did not cause impaired motor function and was not a contributing factor to the observed learning and memory impairments.

Furthermore, translation of the results of animal studies to human studies is difficult, as has been shown for neuroprotection after stroke in the era before reperfusion [39]. In this study healthy adult rats are used, whereas humans often have cardiovascular risk factors such as dyslipidemia and high blood pressure. Moreover, the heterogeneity in humans and the diversity in disease burden is often much larger than that in animals, making results of clinical trials difficult to predict.

In conclusion, adult rats treated *s.c.* with 3 gifts of 2-IB every 12 h in a dose range of 1.1–30 mg/kg/dose directly upon reperfusion showed significant improved memory and learning after 4VO compared to vehicle-treated rats. 2-IB is a suitable candidate for translation to human studies after OHCA.

## Supporting information

**S1 Table. Histology 33 days after four vessel occlusion.** Surviving cells in the CA1 region of the hippocampus at the left and right side for all treatment groups. * p = 0.002 one-way ANOVA, #sham versus all treatment groups p<0.020 Dunnett's post hoc analysis. (DOCX)

## Author Contributions

**Conceptualization:** Cacha Peeters-Scholte, Sigal Meilin.

**Formal analysis:** Sigal Meilin, Paul Westers.

**Investigation:** Yafit Berckovich.

**Methodology:** Yafit Berckovich.

**Project administration:** Sigal Meilin, Yafit Berckovich.

**Resources:** Cacha Peeters-Scholte.

**Supervision:** Sigal Meilin.

**Validation:** Yafit Berckovich.

**Writing – original draft:** Cacha Peeters-Scholte.

**Writing – review & editing:** Cacha Peeters-Scholte.

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
