## [Decision Letter · Decision Letter 0]

5 Jul 2023

PONE-D-22-323872-Iminobiotin, a selective inhibitor of nitric oxide synthase, improves memory and learning in a rat model after four vessel occlusion, mimicking cardiac arrest.PLOS ONE

Dear Dr. Peeters,

Thank you for submitting your manuscript to PLOS ONE. After careful consideration, we feel that it has merit but does not fully meet PLOS ONE’s publication criteria as it currently stands. Therefore, we invite you to submit a revised version of the manuscript that addresses the points raised during the review process.

We look forward to receiving your revised manuscript.

Kind regards,

Giuseppe Pignataro, MD, PhD

Academic Editor

PLOS ONE Journal Requirements:

**PONE-D-22-32387**

3. We note that you have a patent relating to material pertinent to this article. Please provide an amended statement of Competing Interests to declare this patent (with details including name and number), along with any other relevant declarations relating to employment, consultancy, patents, products in development or modified products etc. Please confirm that this does not alter your adherence to all PLOS ONE policies on sharing data and materials, as detailed online in our guide for authors http://journals.plos.org/plosone/s/competing-interests by including the following statement: "This does not alter our adherence to  PLOS ONE policies on sharing data and materials.” If there are restrictions on sharing of data and/or materials, please state these. Please note that we cannot proceed with consideration of your article until this information has been declared.

**Comments to the Author**

1. Is the manuscript technically sound, and do the data support the conclusions?

Reviewer #1: Yes

Reviewer #2: Yes

2. Has the statistical analysis been performed appropriately and rigorously? 

Reviewer #1: I Don't Know

Reviewer #2: Yes

3. Have the authors made all data underlying the findings in their manuscript fully available?

Reviewer #1: Yes

Reviewer #2: No

4. Is the manuscript presented in an intelligible fashion and written in standard English?

Reviewer #1: Yes

Reviewer #2: Yes

5. Review Comments to the Author

Reviewer #1: This is interesting manuscript titled "2-Iminobiotin, a selective inhibitor of nitric oxide synthase, improves memory and learning in a rat model after four vessel occlusion, mimicking cardiac arrest."

The authors should discuss in more detail how selective 2-Iminobiotin is and for what type of nitric oxide synthase.

Reviewer #2: Line 28 - s.c. abbreviation missing.

Line 32 - VO abbreviation missing.

Line 98-100 – References missing in methodology section.

Line 134-135- How are the doses designed and the significance of these dosage regimen. The Pharmacokinetic and pharmacodynamic data about the drug is not mentioned.

Line 141 & 189 - empty lines.

Line 127 – Terminated --- Scarified would be a better and more suitable word.

Line 210 - Parametric References are missing.

Line 230 – Global hypoxic-ischemia – occlusion duration is 15 seconds and the clips on the MCA would be at different time interval. So due to this CA1 regional and histology should be shared and demonstrated even when the difference was not seen.

Line 233 – CA1 region histology missing for 1.1 dosage.

Line 80 – Only female rats are used for this experiment and the relationship between the hormonal cycle is not studied in this study. Hormonal relationship between the 2-IB is missing as well.

Line 310 – Why its is mentioned as 12hrs for 24hrs, the dosage was administered at 12hrs interval for 24hrs.

Line 315 – If the IV bolus dosage of 0.055-0.165mg/kg/dose was demonstrated then why was the regimen set for 10 & 30 mg dose rather the dosage to be considered should be within the +-10-15% variability region. Any justification should be mentioned(missing references).

6. PLOS authors have the option to publish the peer review history of their article (what does this mean?). If published, this will include your full peer review and any attached files.

Reviewer #1: No

Reviewer #2: **Yes: **Rohan Mahesh Patil

---

## [Author Response · Author response to Decision Letter 0]

19 Aug 2023

REBUTTAL LETTER: Response to reviewers Plos One

PONE-D-22-32387

2-Iminobiotin, a selective inhibitor of nitric oxide synthase, improves memory and learning in a rat model after four vessel occlusion, mimicking cardiac arrest.

Thank you very much for carefully reviewing the manuscript and giving the opportunity to improve the manuscript.

Reviewer #1: This is interesting manuscript titled "2-Iminobiotin, a selective inhibitor of nitric oxide synthase, improves memory and learning in a rat model after four vessel occlusion, mimicking cardiac arrest."

The authors should discuss in more detail how selective 2-Iminobiotin is and for what type of nitric oxide synthase.

2-Iminobiotin is an inhibitor of nitric oxide synthase (NOS), but it inhibits the 3 different isoforms (endothelial, neuronal and inducible NOS) in different ways. In in vitro studies, 2-IB, which has a similar chemical structure as L-Arg exhibited selective inhibition of neuronal and inducible nitric oxide synthase activity of murine iNOS and rat cNOS with no effect on endothelial nitric oxide synthase (eNOS) (Sup SJ, 1994). Additionally, radioligand binding assays were performed by Ricerca Biosciences LLC, Taipei, Taiwan for 174 enzymes including all NOS enzymes (Kun-Yuan Lin, 2010). Inhibitory effects of 2-IB were identified against all NOS isoenzymes, but the IC50 was much lower for iNOS and nNOS, than for eNOS, suggesting a more powerful inhibition of inducible and neuronal NOS than endothelial NOS (see Table below).

Biochemical assay

Species Concentration % Inhibition IC50

Nitric oxide synthase, neuronal (nNOS) Rat 300 µM 68 142 µM

Nitric oxide synthase, endothelial (eNOS) Bovine 1000 µM 63 646 µM

Nitric oxide synthase, inducible (iNOS) Mouse 300 µM 81 96 µM

In the discussion lines 307-313 are added to describe the above with the mentioning of 2 references (35,36].

 

Reviewer #2: Line 28 - s.c. abbreviation missing. Is added

Line 32 - VO abbreviation missing. Is written full out now

Line 98-100 – References missing in methodology section. Lines 98-100 are describing in short the methodology, named in reference 14. This was cleared up by adding the words, “in short”.

Line 134-135- How are the doses designed and the significance of these dosage regimen. The Pharmacokinetic and pharmacodynamic data about the drug is not mentioned. We agree with the reviewer that this needs more explanation. Based on earlier studies with 2-iminobiotin in newborn rats, an optimal dose of 10 mg/kg/dose was found, when given in 3 gifts s.c. (directly upon reperfusion and at 12 and 24h after reperfusion). So for this study, we chose a dose regimen with a (3 times) lower and higher treatment schedule to determine the optimal dosage in adult rats after reperfusion. The optimal dosage was added to the methods (ref. 15).

Line 141 & 189 - empty lines. Lines are deleted as well as line 173 and 171

Line 127 – Terminated --- Scarified would be a better and more suitable word. Terminated has been replaced by sacrified.

Line 210 - Parametric References are missing. In sec is added.

Line 230 – Global hypoxic-ischemia – occlusion duration is 15 seconds and the clips on the MCA would be at different time interval. So due to this CA1 regional and histology should be shared and demonstrated even when the difference was not seen. We added a table as Supporting Information File to show the results.

Line 233 – CA1 region histology missing for 1.1 dosage. For a representative example only the best dosage (3.3 mg/kg/dose) was chosen to depict, the 1.1, 10 and 30 mg/kg/dose are not displayed.

Line 80 – Only female rats are used for this experiment and the relationship between the hormonal cycle is not studied in this study. Hormonal relationship between the 2-IB is missing as well. Aim of this study was not to study the hormonal relationship between 2-IB and HI reperfusion injury. But we agree with the reviewer, that this is an important question for translation to clinical studies. This was already described in lines 274-275.

Line 310 – Why its is mentioned as 12hrs for 24hrs, the dosage was administered at 12hrs interval for 24hrs. We agree that this is confusing and changed it to “for 3 gifts” 

Line 315 – If the IV bolus dosage of 0.055-0.165mg/kg/dose was demonstrated then why was the regimen set for 10 & 30 mg dose rather the dosage to be considered should be within the +-10-15% variability region. Any justification should be mentioned(missing references). Currently, the clinical phase2a study is published, which explains the PK rationale. We made a reference to this article [Admiraal MM, Velseboer DC, Tjabbes H, Vis P, Peeters-Scholte C, Horn J. Neuroprotection after cardiac arrest with 2-iminobiotin: a single center phase IIa study on safety, tolerability, and pharmacokinetics. Front Neurol. 2023 Jun 2;14:1136046. doi: 10.3389/fneur.2023.1136046]. 

 

PONE-D-22-32387

As recommended file names were used as provided in the PLOS ONE’s requirements.

We have not received any grants for this research. Neurophyxia B.V. has paid MD Biosciences, the CRO, to perform this trial, which is described in the current manuscript. This is stated in the Financial Disclosures.

3. We note that you have a patent relating to material pertinent to this article. Please provide an amended statement of Competing Interests to declare this patent (with details including name and number), along with any other relevant declarations relating to employment, consultancy, patents, products in development or modified products etc. Please confirm that this does not alter your adherence to all PLOS ONE policies on sharing data and materials, as detailed online in our guide for authors http://journals.plos.org/plosone/s/competing-interests by including the following statement: "This does not alter our adherence to PLOS ONE policies on sharing data and materials.” If there are restrictions on sharing of data and/or materials, please state these. Please note that we cannot proceed with consideration of your article until this information has been declared.

Data are shown now as a Supporting Information file (S1 Table).

In the rebuttal letter all added references are named. No references have been retracted. 

Additional Editor Comments (if provided):

The authors should perform additional experiments to demonstrate how selective 2-Iminobiotin is and for what type of nitric oxide synthase.

Results on selectivity of NOS inhibition are now added to the discussion with reference to 2 earlier studies that investigated the selectivity of NOS [35,36].

---

## [Editor Report · Decision Letter 1]

11 Sep 2023

2-Iminobiotin, a selective inhibitor of nitric oxide synthase, improves memory and learning in a rat model after four vessel occlusion, mimicking cardiac arrest.

PONE-D-22-32387R1

Dear Dr. Peeters,

We’re pleased to inform you that your manuscript has been judged scientifically suitable for publication and will be formally accepted for publication once it meets all outstanding technical requirements.

Kind regards,

Giuseppe Pignataro, MD, PhD

Academic Editor

PLOS ONE
---

## [Editor Report · Acceptance letter]

15 Sep 2023

PONE-D-22-32387R1 

2-Iminobiotin, a selective inhibitor of nitric oxide synthase, improves memory and learning in a rat model after four vessel occlusion, mimicking cardiac arrest. 

Dear Dr. Peeters:

I'm pleased to inform you that your manuscript has been deemed suitable for publication in PLOS ONE. Congratulations! Your manuscript is now with our production department. 

Kind regards, 

on behalf of

Prof. Giuseppe Pignataro 

Academic Editor

PLOS ONE